# Effects of variability of meteorological measures on soil temperature in permafrost regions

Christian Beer<sup>1,2</sup>, Philipp Porada<sup>1,2</sup>, Altug Ekici<sup>1,3</sup>, and Matthias Brakebusch<sup>1,2</sup>

<sup>1</sup>Department of Environmental Science and Analytical Chemistry (ACES), Stockholm University, 10691 Stockholm, Sweden

<sup>2</sup>Bolin Centre for Climate Research, Stockholm University, 10691 Stockholm, Sweden <sup>3</sup>Uni Research Climate, Bjerknes Centre for Climate Research, Bergen, Norway

Correspondence to: Christian Beer (christian.beer@aces.su.se)

**Abstract.** To clarify effects of the variability of meteorological measures and their extreme events on topsoil and subsoil temperature in permafrost regions, an artificially manipulated climate dataset has been used for process-oriented model experiments. Climate variability mainly impacts snow depth, and the cover and thermal diffusivity of lichens and bryophytes. The latter effect is of opposite di-

- rection in summer and winter. These impacts of climate variability on insulating layers together substantially alter the heat exchange between atmosphere and soil. As a result, soil temperature is up to 1 K higher when climate variability is reduced under conserved long-term mean meteorological measures. Climate models project warming of the Arctic region but also increasing climate variability and extreme events. Therefore, our results show that projected future increases in permafrost
- temperature and active-layer thickness will be *less pronounced* in response to climate change when considering dynamic snow and near-surface vegetation modules.

# 1 Introduction

Soil temperature is an important physical measure of a terrestrial ecosystem since it controls many functions of microbes and plants. In permafrost regions, soil temperature also defines the biologically active part of the soil that is thawing in summer (active layer). Large-scale soil temperature is mainly determined by vertical heat conduction. Therefore, soil temperature usually follows an annual sinusoidal cycle of air temperature with a damped oscillation (Campbell and Norman, 1998). That is why the projected large increase in air temperature in the Arctic region over the next 100 years (Ciais et al., 2013) is raising large concerns about the response of soil temperature and hence

permafrost thawing in the Arctic. Indeed, measurements during the last decades already show an

increasing permafrost temperature (Romanovsky et al., 2010) and active-layer thickness (Callaghan et al., 2010) in response to global warming. Also, first modelling results confirm such simple response of increasing future soil temperature and active-layer thickness (Schaefer et al., 2011; Koven et al., 2011; Lawrence et al., 2012; Peng et al., 2016). As a result of increasing soil temperature and

- active-layer thickness, heterotrophic respiration is suggested to increase because of the temperatureresponse of biochemical functions (Arrhenius, 1889; van't Hoff, 1896; Lloyd and Taylor, 1994) and the additional availability of decomposable substrate (Koven et al., 2015) potentially leading to a positive climate-carbon cycle feedback (Zimov et al., 2006; Beer, 2008; Heimann and Reichstein, 2008).
- Meteorological measures, such as air temperature and precipitation will not only change gradually into the future but also their short-term variability and frequency of extreme events is projected to change (Easterling et al., 2000; Rahmstorf and Coumou, 2011; Seneviratne et al., 2012). For instance, for northern high-latitude regions, climate models project an increase of the annual maximum of the daily maximum temperature by 4 K until 2100 (Seneviratne et al., 2012) while annual
- maximal daily precipitation is projected to increase by 20% in these areas until 2100. At the same time, many ecosystem functions respond non-linearly to environmental factors, cf. for instance the temperature-dependence of biochemical functions (Arrhenius, 1889). Therefore, effects of the variability of meteorological measures on mean ecosystem functions can enhance or dampen the effects of gradual changes (Reichstein et al., 2013). That is why there is a strong need to understand these
- effects of climate variability on ecosystem states and functions in addition to gradual changes in order to reliably project future ecosystem state dynamics and climate. In this context, effects of climate variability on soil temperature in permafrost environments have not been studied so far.

Due to the well-known dampening effects of soil insulating layers of snow and near-surface vegetation (Yershov, 1998, pages 361-369) (Goodrich, 1982; Zhang, 2005) and the dampening effect of

- soil itself one would expect no to little effects of air temperature fluctuations on soil temperature, in particular not on subsoil and permafrost temperature. However, air temperature variability will have an impact on snow height indirectly through snow density (Abels, 1892) and also directly when temperature is periodically rising above the melting point. The dependence of conductivity on water and ice content (Campbell and Norman, 1998) additionally complicates the picture because water and
- ice contents themselves are also temperature-dependent. This is in particular also true for the layer of lichens and bryophytes above the soil. This layer dynamically influences the vertical heat conduction depending on the water and ice content (Porada et al., 2016a). In addition, climate variability might impact gross productivity (Beer et al., 2014) and hence the cover of lichens and bryophytes itself.
- Snow manipulation experiments have proven the large *spatial* heterogeneity of soil temperature in cold regions due to snow height heterogeneity (Wipf and Rixen, 2010). Here, we aim to investigate the effects of *temporal* variability of meteorological measures on snow and lichen and bryophyte insulating properties and hence soil temperature in permafrost regions. For this, a land surface model

(LSM) has been run with two distinct climate forcing datasets, one control dataset and one that has identical long-term seasonal averages but reduced day-to-day variability. The LSM has recently been advanced by permafrost-specific processes, and in particular also by a dynamic snow representation

(Ekici, 2015) and by a near-surface vegetation model (Porada et al., 2016a).

# 2 Methods

80

## 2.1 Permafrost advancement of the land surface model

- The Jena Scheme for Biosphere-Atmosphere Coupling in Hamburg (JSBACH) is the land surface scheme for the Max Planck Institute Earth System Model (MPI-ESM) (Raddatz et al., 2007). It runs coupled to the atmosphere inside the ESM or offline forced by observation-based or projected climate input data. This model has recently been advanced by several processes which are particularly important in cold regions (Ekici et al., 2014): coupling of soil hydrology and heat balance via latent heat of fusion and the effects of ice and water on thermal properties, and a snow model for soil insulation. The model simulates the heat balance, soil hydrology and carbon cycle in a 1-D vertical scheme. The version used in this study has been updated from the one used in Ekici et al. (2014) by two additional soil layers for thermal and hydrological processes of 13 and 30 m which lead to a total potential soil profile of 53 m which is constrained for soil hydrological processes by the depth to the bedrock. Another constraint on soil hydrological processes is the potentially available pore
- volume which is reduced by ice content.

In contrast to the model version described in Ekici et al. (2014), here we use a snow module that includes a *dynamic* snow density and snow thermal properties (Ekici, 2015). In this approach, the snow density ( $\rho_{snow}$ ) follows a similar representation as in Verseghy (1991). It is initialized with a minimum value of  $\rho_{min} = 50 kgm^{-3}$ . Then the compaction effect is included as a function of time and a maximum density ( $\rho_{max} = 450 kgm^{-3}$ ) value (Eq. 1),

$$\rho_{snow}^{t+1} = \left(\rho_{snow}^t - \rho_{max}\right) \exp \frac{-0.001 \cdot \Delta t}{3600} + \rho_{max} \tag{1}$$

where  $\Delta t$  is the timestep length of model simulation. Additionally, when there is new snowfall, snow density is updated by taking a weighted average of fresh snow density ( $\rho_{min}$ ) and the calculated snow density value of the previous timestep. Snow density controls snow heat transfer parameters.

Eq. 2 and Eq. 3 show the relationships of volumetric snow heat capacity (c<sub>snow</sub>) and snow heat conductivity (λ<sub>snow</sub>) to snow density following the approach of Abels (1892) and Goodrich (1982). With no previous snow layers, c<sub>snow</sub> is initialized with an average value of 0.52MJm<sup>-3</sup>K<sup>-1</sup> and λ<sub>snow</sub> with 0.1Wm<sup>-1</sup>K<sup>-1</sup>,

$$c_{snow} = c_{ice} \cdot \rho_{snow} \tag{2}$$

where  $c_{ice}$  is the specific heat capacity of ice  $(2106Jkg^{-1}K^{-1})$ , and

$$\lambda_{snow} = 2.9 \cdot 10^{-6} \cdot (\rho_{snow})^2$$

(3)

Another important advancement of the JSBACH model version used here is a dynamic lichen and bryophyte model (Porada et al., 2013, 2016a).

- This model is designed to predict lichen and bryophyte net primary productivity (NPP) in a
  process-based way from available light, surface temperature and water content of lichens and bryophytes.
  Furthermore, it is applicable to estimate various impacts of lichens and bryophytes on biogeochemical cycles (Porada et al., 2016b; Lenton et al., 2016). The model includes a dynamic representation of the surface cover which depends on the balance of growth due to NPP and reduction by disturbance, such as fire.
- The lichen and bryophyte water balance is integrated into the scheme of hydrological fluxes in JSBACH. Most importantly, the lichen and bryophyte layer is fully integrated into the heat conduction scheme and hence also functions as a soil insulating layer (Porada et al., 2016a). Soil insulation depends on the fractional grid cell coverage of the lichen and bryophyte layer as well as on its hydrological status. Thereby, thermal diffusivity of this layer is computed as a function of water, ice and
- air content in the lichen and bryophyte layer. The simulated relations between thermal properties of the lichen and bryophyte layer and water content agree well with field observations. Porada et al. (2016a) provides a more detailed description of the dynamic model version used in this study.

#### 2.2 Forcing data

- The driver of the JSBACH model estimates half-hourly climate forcing data using daily data of
   maximum and minimum air temperature, precipitation, short-wave and long-wave radiation, specific humidity and surface pressure. We are using global data at 0.5 degree spatial resolution following the description in (Beer et al., 2014). The historical data from 1901-1978 came from the WATCH forcing dataset (Weedon et al., 2011), and for the period 1979-2010 ECMWF ERA-Interim reanalysis data (Dee et al., 2011) has been bias-corrected against WATCH forcing data as described in Beer et al.
- (2014) (Piani et al., 2010).

Grid cells are divided into four tiles according to the four most dominant plant functional types of this grid cell (Ekici et al., 2014). This vegetation coverage is assumed to stay constant over the time of simulation. In the model simulations used in this study, we apply new soil parameters. Hydrological parameters have been assigned to each soil texture class following Hagemann and Stacke (2015)

according to the percentage of sand, silt and clay at 1 km sptial resolution as indicated by the Harmonized World Soil Database (FAO/IIASA/ISRIC/ISSCAS/JRC, 2012). Thermal parameters have been estimated as in (Ekici et al., 2014) at the 1 km spatial resolution. Then, averages of 0.5-degree grid cells have been calculated. Soil depth until bedrock follows the map used in Carvalhais et al. (2014) based on Webb et al. (2000).

## 125 2.3 Meteorological forcing data with manipulated variability

Based on the climate data described above (subsequently called CNTL dataset), an additional climate dataset has been developed. This dataset shows reduced day-to-day variability but conserved long-term mean values similar to the method described by Beer et al. (2014). Subsequently, the dataset with reduced variability is called REDVAR dataset. In that data set, the variability of daily values is
reduced by a variance factor of k = 0.25 (see Beer et al. (2014) for details), but the seasonal variability is still conserved. The seasonal variability is represented by an 11-year running average across same dates. Differently from Beer et al. (2014), seasonal means in the REDVAR dataset were exactly preserved by normalization with respect to the CNTL dataset for the annual quarters December-January-February, March-April-May, June-July-August, and September-October-November for each year in-

135 dividually.

#### 2.4 Model experiments

JSBACH has been run for the northern hemisphere during 1901-2010. Prior to this period, state variables have been brought into equilibrium using a spin-up approach. We assume the time period 1901-1930 to be a representative for pre-industrial climatology following (Cramer et al., 1999;

- McGuire et al., 2001). Therefore, randomly selected years from that period have been used. For a proper spin-up of soil physical state variables in permafrost regions, we suggest a 2-step procedure. First, a 50-year model run with the above described randomly selected climate from the period 1901-1930 has been done without considering any freezing and thawing. This first spin-up will bring the soil temperature and water pools in a first equilibrium with climate. This is followed by another 100
- 145 years spin-up with the same climate data but now freezing and thawing is switched on in order to have all pools including soil ice and water content, and soil temperature in equilibrium with climate. For addressing the research question about effects of climate variability on soil temperature in permafrost regions (cf. section 1), artificial model experiments are conducted in this study. In addition to the control model run (CNTL), in one model experiment called REDVAR the land surface
- model has been driven by the artificial climate dataset that represents a reduced short-term (day-today) variability while the long-term (30-a) averages are conserved (section 2.3). Then, differences in long-term (seasonal) averages of simulated snow and lichen and bryophyte properties and ultimately soil temperature can be interpreted due to a difference in variability of meteorological measures.

## 3 Results

# 155 3.1 Climate forcing data comparison

The long-term (30-a) averages of air temperature differ by 0.01 K at maximum (Fig. 1a) or 0.004 % between CNTL and REDVAR. Long-term precipitation averages differ between the datasets by only

2e-8 kg m-2 s-1 at maximum (Fig. 1b) which translates into 0.2 %. In other words, these long-term averages are similar between both datasets.

- In contrast, the variability of meteorological measures at daily resolution in both datasets is substantially different. Although the statistical transformation of measures has been performed at residuals to the mean seasonal cycle (Beer et al. (2014), cf. section 2.3), still the standard deviation of air temperature is up to 1 K lower in the REDVAR dataset (Fig. 2a), or 10 %. Interestingly, there are spatial pattern of variability differences with lower standard deviation differences towards colder
- regions, such as East Siberia and the Canadian High Arctic. REDVAR precipitation results are up to 2e-6 kg m-2 s-2 (6 %) lower than the CNTL precipitation data (Fig. 2b).

In the reduced variability climate dataset, high air temperature values (percentile 99) are 1 to 3.5 K lower (Fig. 3a) in permafrost regions compared to CNTL. The percentile 99 of precipitation is 1 to 7 % lower in REDVAR (Fig. 3b).

## 170 3.2 Effects on snow depth and thermal diffusivity

Fig. 4a shows that snow depth is generally higher under reduced climate variability conditions. In fact, the snow depth difference can be explained by differences in snow water equivalents in the same direction (Fig. 4b). In contrast, a possible lower snow density under these conditions cannot explain the difference in snow depth (Fig. 4c). Snow melt flux differences in autumn between both

- model experiments (Fig. 5) demonstrate clearly that under reduced air temperature variability during the beginning of the snow season, individual snow melt events and hence the total snow melt flux are reduced. Interestingly, there is a spatial gradient of snow water equivalent differences between REDVAR and CNTL experiments along with air temperature, e.g. along a longitudinal gradient in Eurasia and from Quebec to Alaska (Fig. 4b). The same longitudinal gradient can be also seen with
- snow melt flux differences (Fig. 5). The reason for this gradient is that in the higher continental climate in Siberia there are less often melting events during the beginning of the snow season than in Europe. Hence, a reduction in air temperature variability hardly reduces such events further in Siberia compared to Europe.

#### 3.3 Effects on cover and thermal diffusivity of lichens and bryophytes

- Under reduced climate variability JSBACH simulates about 2 to 3 percent higher cover of lichens and bryophytes in some boreal and temperate forest regions in Europe and Northern America (Fig. 6a). Differences are however negligible in most of Siberia and tundra. Due to the effects of climate variability on water and ice saturation, lichen and bryophyte thermal diffusivity differs between the experiments by up to 4 percent in summer and 10 percent in winter (Fig. 6). Clear regional differ-
- ences are visible. In European Russia, South Alaska and the US West Coast, thermal diffusivity is larger during the summer but smaller during the winter under reduced climate variability. In Siberia,

differences in winter are negligible while in summer thermal diffusivity is larger under reduced variability.

# 3.4 Effects on soil temperature

- In contrast to the climate forcing data, the long-term average of both topsoil and subsoil temperature differs between REDVAR and CNTL experiments (Fig. 7a,Fig. 7b). In most regions, soil is warmer when climate variability is reduced, most pronounced in West Siberia, European Russia, and Quebec. In these regions, differences can be up to 1 K. These results and also the spatial pattern are similar between topsoil and subsoil values (Fig. 7a,Fig. 7b). The pattern of topsoil temperature differences
- between the model experiments varies between winter and summer season (Fig. 7c,Fig. 7d). In doing so, the winter topsoil temperature pattern is similar to the annual soil temperature pattern shown in Fig. 7a and Fig. 7b. REDVAR summer temperature results are now, in contrast to winter or annual values, lower than results from the CNTL experiment in many permafrost regions, e.g. in Northern Canada or Northwest Siberia, while in many other places still the topsoil is simulated to be warmer
- when climate variability is reduced, e.g. in Northern Central and East Siberia, European Russian tundra or Alaska (Fig. 7d).

#### 4 Discussion

Climate model projections show increasing variability of meteorological measures and hence increasing frequency of extreme meteorological events (Seneviratne et al., 2012) along with a gradu-

- ally changing climate (change of long-term mean values) (Ciais et al., 2013). Because of the nonlinearity of ecosystem response functions, changing extreme events and climate variability generally can have a higher impact on ecosystem state and function than a gradual change of mean meteorological measures (Reichstein et al., 2013). This study contributes to this overall question from a theoretical point of view with two LSM experiments for which artificially manipulated climate forc-
- ing datasets have been employed. These climate datasets practically do not differ in their long-term (30-a) averages while they show a substantial difference in the short-term (daily) variability (section 3.1). Therefore, differences in simulated state variables and fluxes over 30-year periods will be only due to differences in *variability* of meteorological measures. This contributes to our understanding of the effect of climate variability on soil temperature in permafrost regions. The CNTL experiment
- shows *higher* climate variability than the artificial experimental dataset (REDVAR). In section 3 the difference of the manipulated climate experiment (REDVAR) to CNTL is therefore described. However, for interpreting the results in terms of future ecosystem responses to *increasing* climate variability, we carefully invert the direction of the conclusions in this discussion section.
- For land-atmosphere heat conduction the thermal properties of snow and near-surface vegetation 225 (e.g. mosses and lichens), and their spatial extent and height are of major importance (Yershov,

1998; Gouttevin et al., 2012; Wang et al., 2016). Therefore, in this study we first analyze effects of climate variability on these soil insulating layers for a later explanation of the ultimate effects on soil temperature. Snow generally insulates the soil from changing atmospheric temperature. However, effects are smaller during the melting period in spring because the snow is wet and conductivity

- therefore higher, and more importantly, the soil-to-air gradient in temperature is small. The insulation effect of near-surface vegetation also differs among the seasons because of the high dependence of thermal properties on water and ice contents of lichens and bryophytes. Usually, dry lichens and bryophytes during a continental summer should insulate much more than during wet spring or autumn, or during ice-rich winter time.
- Our theoretical study shows that one major effect of higher climate variability on cold region environments is a lower snow water equivalent (section 3.2) which directly translates into lower snow depth values. The potential alternative explanation for lower snow depth would be a higher snow density. However, the results show exactly the opposite (Fig. 4c). In addition to snow depth, snow thermal properties are also an important factor for heat conduction. However, winter snow
- thermal diffusivity is some percent lower under higher climate variability conditions. Therefore, the net *snow-related* effects of climate variability on soil temperature (section 3.4) are explained by snow depth differences alone.

The reason for these snow water equivalent differences are more often circumstances of melting snow during the beginning of the snow season when day-to-day variability of air temperature 245 is higher (section 3.2). These results also point to an interesting combination of impacts of both changing variability *and* gradually changing mean values on ecosystem states because both changes can lead to pass a threshold value (melting point in this case). These impacts can be seen in section 3.2 when combining temporal climate variability effects on snow water equivalent results (Fig. 4) and snow melt flux results (Fig. 5) with longitudinal pattern of these results towards a continental

climate, which can be interpreted in terms of gradual climate change when substituting space for time. Overall, our findings show that projected higher climate variability in future can lead to lower snow depth which will reduce the soil warming in response to air warming.

In addition to the insulating effect of snow, lichens and bryophytes growing on the ground have a high effect on heat conduction. First, higher climate variability can lead to lower cover of lichens and bryophytes (Fig. 6a). The concave light-response curve of photosynthesis is the main reason for a decreasing productivity under increasing climate variability (Beer et al., 2014). However, highest differences in the cover of lichens and bryophytes between REDVAR and CNTL experiments are about 2 % (Fig. 6a) which is in fact not very large. In contrast, it is interesting to note that when climate variability is higher, moss thermal diffusivity can be substantially *higher in winter* and *lower* 

*in summer* in the same region. This fact points to an important role of near-surface vegetation: it will less insulate from air temperature during winter and more insulate during summer with increasing

climate variability in future. Ultimately, these effects of climate variability on moss thermal diffusivity will reduce the soil warming effect of climate change in future.

- Recent modelling studies suggest a soil temperature increase of 0.02 K per year since 1960 (McGuire et al., 2016) which translates into 2 K in 100 years. Such soil temperature increase has also been projected using the JSBACH model under the RCP4.5 scenario (Ekici, 2015) while under the strong warming scenario RCP8.5, the soil temperature increase might be up to 6 to 8 K (Ekici, 2015). However, the models used in McGuire et al. (2016) and Ekici (2015) usually do not account for dynamic snow and lichen and bryophyte properties as we did in this study. Therefore, soil tem-
- perature differences due to climate variability differences alone (section 3.4) of the same order of magnitude (0.5-1 K) demonstrate that under increasing variability of meteorological measures and increasing extreme events in the Arctic (Seneviratne et al., 2012), the effect of gradual air temperature increase on soil temperature and hence active-layer thickness will be *substantially dampened*. Our results are conservative here because the 99 percentiles of air temperature and precipitation
- from the artificial dataset (REDVAR) differ by only 2-3.5 K (temperature) and 1-7 % (precipitation). These values are at the lower end of the range of climate model projections for the Arctic region until 2100 (Seneviratne et al., 2012).

These dampening effects on an otherwise increasing soil temperature in permafrost regions in future will reduce the positive biogeochemical feedback to climate (Zimov et al., 2006; Beer, 2008;

- Heimann and Reichstein, 2008). Therefore, our study demonstrates the need of representing dynamically snow and lichen and bryophyte processes in Earth System Models for a more reliable projection of future soil temperature dynamics and related atmosphere-land exchanges of heat, water and greenhouse gases. A first run of the MPI-ESM with the permafrost-advanced land surface scheme JSBACH coupled to the atmosphere model showed a remarkable bias in 2m air tempera-
- ture of 1-4 K in permafrost regions compared to the standard model version without freezing and thawing (Hagemann et al., 2016). Our results show that this bias could be potentially reduced when implementing dynamic snow and lichen and bryophyte modules.

### 5 Conclusions

Artificial model experiments show an impact of the variability of meteorological measures and their
extreme events on the long-term mean of soil temperature in permafrost-affected terrestrial ecosystems. This impact is due to temperature variability effects on snow melt hence snow depth as well as climate variability effects on the cover and (seasonally different) thermal diffusivity of lichens and bryophytes. Overall, the soil temperature response to increasing climate variability and extreme events (soil cooling) will be opposite to the response of soil temperature to gradually increasing air temperature (soil warming). This shows the importance of representing dynamically snow and

lichen and bryophyte functions in Earth System Models for projecting future permafrost soil states and land-atmosphere interactions, hence future climate.

Acknowledgements. Financial support came from the European Union FP7-ENV project PAGE21 under contract number GA282700. Model simulations were performed on resources provided by the Swedish National

Infrastructure for Computing (SNIC) at Linköping University. We acknowledge the Land Department, Max Planck Institute for Meteorology, Hamburg, Germany for JSBACH code maintenance.

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
