# Peer review of "Effects of variability of meteorological measures on soil temperature in permafrost regions"

_The Cryosphere, 2016_

## Referee Comment (RC1) · Anonymous Referee #1 · 27 Sep 2016

The current version of the paper requires significant modifications and rewrites in order to be considered for the Cryosphere journal. I think that this paper is more suitable for Geoscientific Model Development journal and suggest to authors to consider that journal for this paper. Overall, it looks to me that authors promote usage of the dynamics snow and organic layer in the process-oriented models, which sounds more like advances in the model that authors employed in their work.

Authors state that climate variability mostly impacts snow depth and the upper soil organic layer (SOL). Authors call SOL as cover and thermal diffusivity of lichen and bryophytes. These different notations confuse the reader right from the beginning. First describe the lichen and bryophytes. Do not assume that readers know everything about them. Spend more lines on the description in the introduction and methods sections. What are their spatial coverage and thickness? Why are they so important?

Note, that SOL thickness and the level of saturation will determine soil temperature for the certain location and time.

In Abstract. Where is the 1K higher temperature come from? Is this temperature difference uniform for every geographic location? Statements like 'less-pronounced' in the abstract looks like a hand-waving to me. Please use exact numbers (statistics) when making any statements in the manuscript.

I suggest to review the corresponding literature and cite previous work appropriately in this study. For example, recent work summarizing the models inter-comparison on modeling of snow (Wang et al., 2016). Recent work stressing the importance of the organic layer and coupling of the soil biogeochemical processes in the land system models (Jafarov et al., 2016). Once again changes in the SOL heat diffusion properties directly correlate with the level of saturation of soil for a specific year (see O'Donnell et al., 2009 and many others).

In conclusion, there are improvement that has to be done through the entire paper. If resubmitted for the Cryosphere journal then the advancement in science has to be better stressed.

References

1. O'Donnell, Jonathan A.; Romanovsky, Vladimir E.; Harden, Jennifer W.; McGuire, A. David. 2009. The effect of moisture content on the thermal conductivity of moss and organic soil horizons from black spruce ecosystems in interior Alaska. Soil Science. 174(12): 646-651.

2. Jafarov, E. and Schaefer, K.: The importance of a surface organic layer in simulating permafrost thermal and carbon dynamics, The Cryosphere, 10, 465-475, doi:10.5194/tc-10-465-2016, 2016

3 Wang, W., Rinke, A., Moore, J. C., Ji, D., Cui, X., Peng, S., Lawrence, D. M., McGuire, A. D., Burke, E. J., Chen, X., Decharme, B., Koven, C., MacDougall, A.,

Saito, K., Zhang, W., Alkama, R., Bohn, T. J., Ciais, P., Delire, C., Gouttevin, I., Hajima, T., Krinner, G., Lettenmaier, D. P., Miller, P. A., Smith, B., Sueyoshi, T., and Sherstiukov, A. B.: Evaluation of air–soil temperature relationships simulated by land surface models during winter across the permafrost region, The Cryosphere, 10, 1721-1737, doi:10.5194/tc-10-1721-2016, 2016.

---

## Referee Comment (RC2) · C. Koven (Referee) · 3 Oct 2016

This paper presents an exploration of a fairly theoretical question: what role does high frequency variability in meteorology have on the low-frequency dynamics of soil temperature in arctic regions. The authors show that, because snow tends to melt during brief warm period in the fall and winter, when temperature variability is reduced, less snow melts, which leads to more snow accumulation and warmer soils. This is not immediately obvious given the expectation that soil acts as a strong low band pass filter on temperature variability, but once explained, it makes perfect sense. Conversely, there is also a mechanism associated with bryophytes, which is less well explained in the text, but which leads to somewhat the opposite response.

I have a couple issues with the paper, but I first want to say that I disagree with the first

reviewer's comment that this is more appropriate for GMD. This is clearly a model application rather than a model development paper, so I think this journal is an appropriate one for the work.

One issue I have is in the explanation for the bryophyte mechanism. I get the argument about reduced productivity leading to reduced bryophyte cover, but this appears minor in the actual permafrost region itself. So I don't understand why the effects on soil physical properties ought to be large. Is this just an outcome of the soils being warmer and therefore thawed longer in the spring and fall or deeper in the summer under reduced variability, which then leads to different time-averaged soil physical properties since the thawed and frozen states are so different for organic soils? If this is the mechanism, then it is really specific to the bryophyte representation per se, or ought any realistic organic soil parameterization give qualitatively the same result? In any case, the fact that soil temperatures in the annual mean appear to be almost entirely warmer in the reduced variability case argues that the snow mechanism is the dominant one, so you probably ought to state as much in the paper.

A second issue I have is in what these idealized results actually mean in terms of more policy-relevant questions such as climate projections. Under warming, snow cover is expected to be reduced over most of the warmer part of the permafrost region. So that ought to attenuate the importance of the snow mechanism here. Furthermore, models which are driven directly by GCM future scenarios ought to explicitly capture this effect, so in principle this may already be built into those permafrost projections. But perhaps an issue is that for some studies, e.g. those which are used in the permafrost carbon network future scenario MIP, and many papers that are using that output, (and which I've been involved in, hence my waiving of anonymity in this review), the protocol forcing data for the future period is a hybrid that uses high frequency data that comes from reanalysis data, but with climatological anomalies relative to the present from a GCM imposed onto the reanalysis output for future. This design was chosen to avoid the mean-state biases present in almost all GCMs, but perhaps it leads to new

problems that are exposed here if the variance changes. I.e., such an experimental design implicitly assumes that only the long-term means are changing and not the high frequency variability about the means, which is still coming from the reanalysis data. So, how big an effect is there from ignoring the projected changes in the variability? I suggest this paper would be most useful to the community if it actually tried to answer this question. Doing so properly would require a few more runs: i.e. something along the lines of taking a GCM scenario meteorology output, and then either include changes to the variance over time or not, while in both cases leaving the changes in the mean state due to the scenario, and then use those to force JSBACH offline to assess how large the bias in projected soil temperature, permafrost extent, etc, changes are in the current PCN-MIP way things are done. I recognize this would require substantially more work on the authors' part, but I'd suggest the authors consider doing such an experiment if they really feel like this is an important result that the community needs to take seriously.

A last minor issue, on page 9, line 268. This statement needs a lot more support if it is to remain. Several of the models in that study have some version of the effects described here. While every model that does will include effects like organic soil and snow physics differently, it is not at all clear that most of the models in that MIP will not include the essence of these effects. I don't think it is the authors' intention to assert that no other models have snow which melts during brief warm spells, or that the models that include organic soils (but not bryophytes) necessarily miss out on all of those effects described here, so I'd suggest not making the statement.

Minor point: The symbol (-) as a unit in several figures is unclear. I see that you are using it to mean relative fractional differences, but please state that somewhere in each figure.

---

## Author Comment (AC1) · 10 Oct 2016

The current version of the paper requires significant modifications and rewrites in order to be considered for the Cryosphere journal. I think that this paper is more suitable for Geoscientific Model Development journal and suggest to authors to consider that journal for this paper. Overall, it looks to me that authors promote usage of the dynamics snow and organic layer in the process-oriented models, which sounds more like advances in the model that authors employed in their work.

We like to thank the reviewer for taking the time to read and review this manuscript and for a fast comment. Advancements of the previous manuscript are highlighted in red in the revised version of the manuscript.

As is written in the abstract and the last paragraph of the introduction section, and also explained in detail in the first paragraph of the discussion section, this paper has a clear research question: What are effects of short-term variability of meteorological measures on soil temperature in permafrost regions? The research gap behind is explained in the second and third paragraphs of the introduction section.

In general one can ask this question for any other ecosystem state or function, and a whole EU FP7 Project (www.carbo-extreme.eu) was funded to find answers, mainly on the carbon cycle, cf. Reichstein et al. (2013). There are several different approaches possible to answer such question, e.g. one could design lab or field experiments (cf. CARBO-Extreme project) in which the climate forcing to the ecosystem is manipulated. That would be really interesting to do also for the specific question here about effects on permafrost soil temperature, see also new discussion text lines 363-364. However, in reality the experimental design is very complicated (because the averages need to be conserved), very expensive, would be required at several locations, and, when looking at subsoil temperature, required also long periods. Therefore, we try to give first answers from a theoretical point of view, mimicking reality with a process-oriented land surface scheme forced by artificially modified climate forcing data. In doing so, it is for example much easier to keep the long-term average equal to the control, and we can see results at a continental scale including a huge range of baseline environmental conditions (climate regimes and soil properties).

The model used to address this question has already been developed and published:
Porada, P.; Ekici, A. & Beer, C. (2016) Effects of bryophyte and lichen cover on permafrost soil temperature at large scale, The Cryosphere 10, 2291-2315.

Therefore, we think that The Cryosphere is the right journal for addressing this research question.

Authors state that climate variability mostly impacts snow depth and the upper soil organic layer (SOL). Authors call SOL as cover and thermal diffusivity of lichen and bryophytes. These different notations confuse the reader right from the beginning. First describe the lichen and bryophytes. Do not assume that readers know everything about them. Spend more lines on the description in the introduction and methods sections. What are their spatial coverage and thickness? Why are they so important? Note, that SOL thickness and the level of saturation will determine soil temperature for the certain location and time.

We fully agree with the reviewer, and it is also see by the model experiments, that the properties of this near-surface vegetation is of major importance for soil temperature. Mosses and lichens are growing on top of any soil organic layer and respond dynamically to climate and CO2. That was also the reason why, in a major **previous** effort, a dynamic vegetation model of lichens and bryophytes has been included into the JSBACH land surface scheme (Porada et al., 2016). When applying this new "moss model" in the context of the present research question, we see the importance of this near-surface vegetation layer.

Wording and definitions, however, should be clear from the beginning. Most of the new "moss model" functions are described in the last paragraphs of section 2.1 and the reader is of course also referred to the Porada et al. (2016) paper in The Cryosphere which is also open access. Still we agree on taking more space in the introduction to clarify what is meant by lichens and bryophytes. For this, we introduce a new fourth paragraph to the introduction section, lines 54-64.

In Abstract. Where is the 1K higher temperature come from? Is this temperature difference uniform for every geographic location? Statements like 'less-pronounced' in the abstract looks like a hand-waving to me. Please use exact numbers (statistics) when making any statements in the manuscript.

This is the main result of the REDVAR experiment as shown in Fig 7 and described in section 3.4. In the revised version, we advance the sentence to the following:
"As a result, soil temperature is 0.2 to 1 K higher when climate variability is reduced under conserved long-term mean meteorological measures, depending on the location."

The last sentence of the abstract is to recall the overall conclusion of the paper as it has been discussed in the discussion section. In the revised version of the manuscript this sentence now reads as "Therefore, our results show that projected future increases in permafrost temperature and active-layer thickness in response to climate change will be lower i) when taking into account future changes in short-term variability of meteorological measures, and ii) when representing dynamic snow and lichen and bryophyte functions in land surface models."

I suggest to review the corresponding literature and cite previous work appropriately in this study. For example, recent work summarizing the models inter-comparison on modeling of snow (Wang et al., 2016). Recent work stressing the importance of the organic layer and coupling of the soil biogeochemical processes in the land system models (Jafarov et al., 2016). Once again changes in the SOL heat diffusion properties directly correlate with the level of saturation of soil for a specific year (see O'Donnell et al., 2009 and many others).

We thank the reviewer for these suggestions and advanced citations in the introduction and discussion sections. However, we would also like to point out that the importance of snow and soil properties on heat conduction as well as their dependence on environmental conditions has been studied and reported extensively during the past century and that it is not our aim to give a balanced amount of citations on that fact. For the research question of this paper it is more important to remind the reader on these functions and we think it would be more useful to give citations to textbooks for readers that want to learn about it.

In conclusion, there are improvement that has to be done through the entire paper. If resubmitted for the Cryosphere journal then the advancement in science has to be better stressed.

We hope that we could demonstrate the advancement in science that merits a publication in The Cryosphere by addressing a specific important research question.

References
1. O'Donnell, Jonathan A.; Romanovsky, Vladimir E.; Harden, Jennifer W.; McGuire, A. David. 2009. The effect of moisture content on the thermal conductivity of moss and organic soil horizons from black spruce ecosystems in interior Alaska. Soil Science. 174(12): 646-651.
2.Jafarov, E. and Schaefer, K.: The importance of a surface organic layer in simulating permafrost thermal and carbon dynamics, The Cryosphere, 10, 465-475, doi:10.5194/tc-10-465-2016, 2016
3 Wang, W., Rinke, A., Moore, J. C., Ji, D., Cui, X., Peng, S., Lawrence, D. M., McGuire, A. D., Burke, E. J., Chen, X., Decharme, B., Koven, C., MacDougall, A., Saito, K., Zhang, W., Alkama, R., Bohn, T. J., Ciais, P., Delire, C., Gouttevin, I., Hajima, T., Krinner, G., Lettenmaier, D. P., Miller, P. A., Smith, B., Sueyoshi, T., and Sherstiukov, A. B.: Evaluation of air–soil temperature relationships simulated by land surface models during winter across the permafrost region, The Cryosphere, 10, 1721-1737, doi:10.5194/tc-10-1721-2016, 2016

---

## Author Comment (AC2) · 10 Oct 2016

C. Koven (Referee)

This paper presents an exploration of a fairly theoretical question: what role does high frequency variability in meteorology have on the low-frequency dynamics of soil temperature in arctic regions.  The authors show that, because snow tends to melt during brief warm period in the fall and winter, when temperature variability is reduced, less snow melts,  which leads to more snow accumulation and warmer soils.  This is not immediately obvious given the expectation that soil acts as a strong low band pass filter on temperature variability, but once explained, it makes perfect sense. Conversely, there is also a mechanism associated with bryophytes, which is less well explained in the text, but which leads to somewhat the opposite response.

We like to thank you for a constructive review that helped to improve the manuscript. Advancements of the previous manuscript are highlighted in red in the revised version of the manuscript.

I have a couple issues with the paper, but I first want to say that I disagree with the first reviewer's comment that this is more appropriate for GMD. This is clearly a model application rather than a model development paper, so I think this journal is an appropriate one for the work.

One issue I have is in the explanation for the bryophyte mechanism. I get the argument about reduced productivity leading to reduced bryophyte cover, but this appears minor in the actual permafrost region itself.   So I don't understand why the effects on soil physical properties ought to be large. Is this just an outcome of the soils being warmer and therefore thawed longer in the spring and fall or deeper in the summer under reduced variability, which then leads to different time-averaged soil physical properties since the thawed and frozen states are so different for organic soils?  If this is the mechanism, then it is really specific to the bryophyte representation per se, or ought any realistic organic soil parameterization give qualitatively the same result?  In any case, the fact that soil temperatures in the annual mean appear to be almost entirely warmer in the reduced variability case argues that the snow mechanism is the dominant one, so you probably ought to state as much in the paper.

The mosses and lichens layer that has been implemented into the LSM (Porada et al., TC, 2016) is a dynamic vegetation model representing near-surface vegetation, in contrast to a soil organic layer. Its cover will change in response to CO2 and climate changes in future. Importantly, this near-surface vegetation also functions as an additional thermal and hydrological layer; it represents water and ice stocks and is part of the energy balance and hydrological scheme. The general impacts of such near-surface vegetation are described in the third paragraph of the introduction. Main functions are the vegetation cover, and its water and ice content. The latter two determine the thermal conductivity and heat capacity and hence thermal diffusivity. In the introduction section lines 54-64 and methods section lines 103-119 and also Porada et al (2016) the functionality of the lichens and bryophyte layer is described more in detail.

Interestingly, in most regions thermal diffusivity of the moss layer is higher in summer and lower in winter when climate variability is reduced (Fig 6, section 3.3). The reason is the impact of climate variability on moss water and ice contents. These effects of climate variability on moss thermal

properties have effects on soil temperature in the same direction than snow depth, it makes the soil warmer. To make this clear, we add the following sentence to results and discussion sections: "These effects of climate variability on thermal diffusivity of lichens and bryophytes and hence soil temperature are in the same direction as effects."

Note, that in the discussion section, however, we discuss the opposite situation of increase instead of decreasing variability, which will consequently lead to cooler soils. Still, snow and moss effects go in the same direction. Hence, the important effects of climate variability on seasonal moss thermal diffusivity are discussed in lines 308-320.

Your question if such effects can be represented by a soil organic layer parametrization is interesting from a model development point of view. The wish would be to treat mosses and lichens as part of the organic soil horizon because that is sometimes already implemented in LSMs. In reality, mosses and lichens grow on top of an organic layer and their hydrological regimes differ. The vegetation functions will respond faster to environmental change, and hydrological and thermal properties of lichens and bryophytes differ from soil organic layer properties. For example mosses have a higher porosity and therefore can store more water and ice, but potentially also dry faster in summer. Most importantly, we represent moss cover at the sub-grid scale while the organic horizon usually is represented uniform for a certain grid cell. Still, specific model tests with a version including an organic layer that is fully part of water and energy balances (we do not have such representation), and a moss model like Porada et al. (2016) would be required in order to quantify the net difference and to fully answer your question. Such study would be really interesting for a model intercomparison paper about heat conduction representations but is clearly out of the scope of this paper.

A second issue I have is in what these idealized results actually mean in terms of more policy-relevant questions such as climate projections. Under warming, snow cover is expected to be reduced over most of the warmer part of the permafrost region. So that ought to attenuate the importance of the snow mechanism here. Furthermore, models which are driven directly by GCM future scenarios ought to explicitly capture this effect, so in principle this may already be built into those permafrost projections. But perhaps an issue is that for some studies, e.g. those which are used in the permafrost carbon network future scenario MIP, and many papers that are using that output, (and which I've been involved in, hence my waiving of anonymity in this review), the protocol forcing data for the future period is a hybrid that uses high frequency data that comes from reanalysis data, but with climatological anomalies relative to the present from a GCM imposed onto the reanalysis output for future. This design was chosen to avoid the mean-state biases present in almost all GCMs, but perhaps it leads to new problems that are exposed here if the variance changes. I.e., such an experimental design implicitly assumes that only the long-term means are changing and not the high frequency variability about the means, which is still coming from the reanalysis data. So, how big an effect is there from ignoring the projected changes in the variability? I suggest this paper would be most useful to the community if it actually tried to answer this question. Doing so properly would require a few more runs: i.e. something along the lines of taking a GCM scenario meteorology output, and then either include changes to the variance over time or not, while in both cases leaving the changes in the mean state due to the scenario, and then use those to force JSBACH offline to assess how large the bias in projected soil temperature, permafrost extent, etc, changes are in the current PCN-MIP way things are done. I recognize this would require substantially

more work on the authors' part, but I'd suggest the authors consider doing such an experiment if they really feel like this is an important result that the community needs to take seriously.

Thank you for this very important point. Actually, since raw ESM output data will be biased we suggest avoiding JSBACH experiments forced by this original climate model outputs. The specific numbers of equilibrium state variables (from model spin-up), such as soil temperature, are important for quantifying any effects due to artificial model experiments. Still, we see a way of directly quantifying this impact of continuously changing short-term variability of meteorological measures into the future. Forcing data for the CNTL run would be from a state-of-the-art statistical bias correction of ESM output data, and for the additional artificial experiment the short-term variability should be dynamically reduced in time until 2100 (Fig 8a,b).

For the revised version of the paper, we extend the study substantially by
i) creating such climate forcing data and
ii) run such additional experiment (REDVARfut) for a single representative region at 70N/100E
in order to demonstrate that by using such slightly different method we will also arrive at the same conclusions: Increasing variability of meteorological measures in future will lead to less pronounced warming of the permafrost soil in response to climate change. For this, methods, results and discussion sections have been substantially advanced (red color). In particular, the reviewer comment and the new experiment also advanced the discussion section: Our findings also show that climate data preparation should continuously be improved also w.r.t. higher statistical moments and a change in short-term variability. This has been also added into the discussion section lines 360-362.

Conclusions and the last sentence of the abstract have been also revised in order to highlight this discussion about future forcing data preparation and the new results.

A last minor issue, on page 9, line 268. This statement needs a lot more support if it is to remain. Several of the models in that study have some version of the effects described here. While every model that does will include effects like organic soil and snow physics differently, it is not at all clear that most of the models in that MIP will not include the essence of these effects. I don't think it is the authors' intention to assert that no other models have snow which melts during brief warm spells, or that the models that include organic soils (but not bryophytes) necessarily miss out on all of those effects described here, so I'd suggest not making the statement.

We fully agree with you and deleted that sentence. It is not the aim of this study to compare different process-oriented models in the way they represent heat conduction.

Minor point: The symbol (-) as a unit in several figures is unclear. I see that you are using it to mean relative fractional differences, but please state that somewhere in each figure

Hope it is more clear now.

---

## Author Comment (AC3) · 12 Oct 2016

Manuscript prepared for The Cryosphere
with version 2014/09/16 7.15 Copernicus papers of the LaTeX class copernicus.cls.
Date: 10 October 2016

**Effects of variability of meteorological measures on soil temperature in permafrost regions**

Christian Beer[1,2], Philipp Porada[1,2], Altug Ekici[1,3], and Matthias Brakebusch[1,2]

[1]Department of Environmental Science and Analytical Chemistry (ACES), Stockholm University, 10691 Stockholm, Sweden
[2]Bolin Centre for Climate Research, Stockholm University, 10691 Stockholm, Sweden
[3]Uni Research Climate, Bjerknes Centre for Climate Research, Bergen, Norway

*Correspondence to:* Christian Beer (christian.beer@aces.su.se)

**Abstract.** To clarify effects of the variability of meteorological measures and their extreme events on topsoil and subsoil temperature in permafrost regions, an artificially manipulated climate dataset has been used for process-oriented model experiments. Climate variability mainly impacts snow depth, and the cover and thermal diffusivity of lichens and bryophytes. The latter effect is of opposite direction in summer and winter. These impacts of climate variability on insulating layers together substantially alter the heat exchange between atmosphere and soil. As a result, soil temperature is 0.2 to 1 K higher when climate variability is reduced under conserved long-term mean meteorological measures, depending on the location. Earth system models project warming of the Arctic region but also increasing variability of meteorological measures and extreme events. Therefore, our results show that projected future increases in permafrost temperature and active-layer thickness in response to climate change will be lower i) when taking into account future changes in short-term variability of meteorological measures, and ii) when representing dynamic snow and lichen and bryophyte functions in land surface models.

**1 Introduction**

Soil temperature is an important physical measure of a terrestrial ecosystem since it controls many functions of microbes and plants. In permafrost regions, soil temperature also defines the biologically active part of the soil that is thawing in summer (active layer). Large-scale soil temperature is mainly determined by vertical heat conduction. Therefore, soil temperature usually follows an annual sinusoidal cycle of air temperature with a damped oscillation (Campbell and Norman, 1998). That is why the projected large increase in air temperature in the Arctic region over the next 100 years (Ciais et al., 2013) is raising large concerns about the response of soil temperature and hence

permafrost thawing in the Arctic. Indeed, measurements during the last decades already show an increasing permafrost temperature (Romanovsky et al., 2010) and active-layer thickness (Callaghan et al., 2010) in response to global warming. Also, first modelling results confirm such simple re-

25     sponse of increasing future soil temperature and active-layer thickness (Schaefer et al., 2011; Koven et al., 2011; Lawrence et al., 2012; Peng et al., 2016). As a result of increasing soil temperature and active-layer thickness, heterotrophic respiration is suggested to increase because of the temperature-response of biochemical functions (Arrhenius, 1889; van't Hoff, 1896; Lloyd and Taylor, 1994) and the additional availability of decomposable substrate (Koven et al., 2015) potentially leading to a

30     positive climate-carbon cycle feedback (Zimov et al., 2006; Beer, 2008; Heimann and Reichstein, 2008).

Meteorological measures, such as air temperature and precipitation will not only change gradually into the future but also their short-term variability and frequency of extreme events is projected to change (Easterling et al., 2000; Rahmstorf and Coumou, 2011; Seneviratne et al., 2012). For

35     instance, for northern high-latitude regions, climate models project an increase of the annual maximum of the daily maximum temperature by 4 K until 2100 (Seneviratne et al., 2012) while annual maximal daily precipitation is projected to increase by 20% in these areas until 2100. At the same time, many ecosystem functions respond non-linearly to environmental factors, cf. for instance the temperature-dependence of biochemical functions (Arrhenius, 1889). Therefore, effects of the vari-

40     ability of meteorological measures on mean ecosystem functions can enhance or dampen the effects of gradual changes (Reichstein et al., 2013). That is why there is a strong need to understand these effects of climate variability on ecosystem states and functions in addition to gradual changes in order to reliably project future ecosystem state dynamics and climate. In this context, effects of climate variability on soil temperature in permafrost environments have not been studied so far.

45     Due to the well-known dampening effects of soil insulating layers of snow and near-surface vegetation (Yershov, 1998, pages 361-369) (Goodrich, 1982; Zhang, 2005; Wang et al., 2016) and the dampening effect of the soil organic layer (Jafarov and Schaefer, 2016) and mineral soil (Campbell and Norman, 1998) itself one would expect no to little effects of changing air temperature fluctuations on soil temperature, in particular not on subsoil and permafrost temperature. However, air

50     temperature variability will have an impact on snow height indirectly through snow density (Abels, 1892) and also directly when temperature is periodically rising above the melting point. The dependence of conductivity on water and ice content (Campbell and Norman, 1998) additionally complicates the picture because water and ice contents themselves are also temperature-dependent.

Lichens are symbiotic organisms consisting of a fungus and at least one green alga or cyanobac-

55     terium, while bryophytes are non-vascular plants which have no specialised tissue such as roots or stems. Both groups cannot actively control their water uptake and loss, but they tolerate drying and are able to reactivate their metabolism on rewetting. Typical species of upland regions at high latitudes are feather mosses such as *Hylocomium splendens* and *Pleurozium schreberi* or the lichen

*Cladonia stellaris*. This near-surface vegetation is growing on top of any organic horizon and hence important for heat fluxes between land and atmosphere. In particular also for this layer, thermal and hydrological properties depend highly on water and ice content. Hence, lichens and bryophytes dynamically influence the vertical heat conduction (Porada et al., 2016a). In addition, climate variability might impact gross productivity (Beer et al., 2014) and hence the cover of lichens and bryophytes itself, which ranges between 0 and 100 % depending on the environmental conditions.

[revised manuscript text omitted]

These effects of climate variability on thermal diffusivity of lichens and bryophytes and hence soil temperature are in the same direction as snow effects (section 3.2).

**3.4 Effects on soil temperature**

In contrast to the climate forcing data, the long-term average of both topsoil and subsoil temperature differs between REDVAR and CNTL experiments (Fig. 7a,Fig. 7b). In most regions, soil is warmer when climate variability is reduced, most pronounced in West Siberia, European Russia, and Quebec. In these regions, differences can be up to 1 K. These results and also the spatial pattern are similar between topsoil and subsoil values (Fig. 7a,Fig. 7b). The pattern of topsoil temperature differences between the model experiments varies between winter and summer season (Fig. 7c,Fig. 7d). In doing so, the winter topsoil temperature pattern is similar to the annual soil temperature pattern shown in Fig. 7a and Fig. 7b. REDVAR summer temperature results are now, in contrast to winter or annual values, lower than results from the CNTL experiment in many permafrost regions, e.g. in Northern Canada or Northwest Siberia, while in many other places still the topsoil is simulated to be warmer when climate variability is reduced, e.g. in Northern Central and East Siberia, European Russian tundra or Alaska (Fig. 7d).

**3.5 Effects of future changes of climate variability on soil temperature**

In order to analyze effects of changing variability of meteorological measures in time until 2100, the results of the specific model runs into the future at 70N/100E are displayed as time series in Fig. 8. The bias-corrected MPI-ESM CMIP5 model output following RCP8.5 shows increasing air temperature (Fig. 8a) and increasing precipitation (Fig. 8b). This positive trend is also seen by the annual minimum (percentile 1) and maximum (percentile 99) temperature (Fig. 8a) and precipitation (Fig. 8b). Meteorological forcing data of the REDVARfut dataset shows similar long-term averages to the CNTL dataset (Fig. 8a,Fig. 8b). Hence, REDVARfut variables follow the general positive trend. However, as the short-term variability is designed to being increasingly reduced, the differences in the minima and maxima air temperature are increasing during 2011-2100 (Fig. 8a). The increasing precipitation at the high end of the distribution in the CNTL dataset has been reversed in REDVARfut where the amount of precipitation at percentile 99 is decreasing in time (Fig. 8b).

These time-varying changes in the variability of meteorological measures under conserved long-term average have effects on topsoil temperature of up to 0.9 K (Fig. 8c), i.e. the overall increasing topsoil temperature due to increasing air temperature is a bit higher in case of reduced climate variability. We even see some small effect in 38 m depth (Fig. 8d) even though short-term atmospheric data fluctuations should be most filtered at this depth.

**4  Discussion**

Climate model projections show increasing variability of meteorological measures and hence increasing frequency of extreme meteorological events (Seneviratne et al., 2012) along with a gradually changing climate (change of long-term mean values) (Ciais et al., 2013). Because of the nonlinearity of ecosystem response functions, changing extreme events and climate variability generally can have a higher impact on ecosystem state and function than a gradual change of mean meteorological measures (Reichstein et al., 2013). This study contributes to this overall question from a theoretical point of view with two LSM experiments for which artificially manipulated climate forcing datasets have been employed. These climate datasets practically do not differ in their long-term (30-a) averages while they show a substantial difference in the short-term (daily) variability (section 3.1). Therefore, differences in simulated state variables and fluxes over 30-year periods will be only due to differences in *variability* of meteorological measures. This contributes to our understanding of the effect of climate variability on soil temperature in permafrost regions. The CNTL experiment shows *higher* climate variability than the artificial experimental dataset (REDVAR). In section 3 the difference of the manipulated climate experiment (REDVAR) to CNTL is therefore described. However, for interpreting the results in terms of future ecosystem responses to *increasing* climate variability, we carefully invert the direction of the conclusions in this discussion section.

For land-atmosphere heat conduction the thermal properties of snow, near-surface vegetation (e.g. mosses and lichens), and the soil organic layer, and their spatial extent and height are of major importance (Yershov, 1998; Gouttevin et al., 2012; Wang et al., 2016; Jafarov and Schaefer, 2016). Therefore, in this study we first analyze effects of climate variability on these soil insulating layers for a later explanation of the ultimate effects on soil temperature. Snow generally insulates the soil from changing atmospheric temperature. However, effects are smaller during the melting period in spring because the snow is wet and conductivity therefore higher, and more importantly, the soil-to-air gradient in temperature is small. The insulation effect of near-surface vegetation also differs among the seasons because of the high dependence of thermal properties on water and ice contents of lichens and bryophytes. Usually, dry lichens and bryophytes during a continental summer should insulate much more than during wet spring or autumn, or during ice-rich winter time.

Our theoretical study shows that one major effect of higher climate variability on cold region environments is a lower snow water equivalent (section 3.2) which directly translates into lower snow depth values. The potential alternative explanation for lower snow depth would be a higher snow density. However, the results show exactly the opposite (Fig. 4c). In addition to snow depth, snow thermal properties are also an important factor for heat conduction. However, winter snow thermal diffusivity is some percent lower under higher climate variability conditions. Therefore, the net *snow-related* effects of climate variability on soil temperature (section 3.4) are explained by snow depth differences alone.

The reason for these snow water equivalent differences are more often circumstances of melting snow during the beginning of the snow season when day-to-day variability of air temperature is higher (section 3.2). These results also point to an interesting combination of impacts of both changing variability *and* gradually changing mean values on ecosystem states because both changes can lead to pass a threshold value (melting point in this case). These impacts can be seen in section 3.2 when combining temporal climate variability effects on snow water equivalent results (Fig. 4) and snow melt flux results (Fig. 5) with longitudinal pattern of these results towards a continental climate, which can be interpreted in terms of gradual climate change when substituting space for time. Overall, our findings show that projected higher climate variability in future can lead to lower snow depth which will reduce the soil warming in response to air warming.

In addition to the insulating effect of snow, lichens and bryophytes growing on the ground have a high effect on heat conduction. First, higher climate variability can lead to lower cover of lichens and bryophytes (Fig. 6a). The concave light-response curve of photosynthesis is the main reason for a decreasing productivity under increasing climate variability (Beer et al., 2014). However, highest differences in the cover of lichens and bryophytes between REDVAR and CNTL experiments are about 2 % (Fig. 6a) which is in fact not very large. In contrast, it is interesting to note that when climate variability is higher, moss thermal diffusivity can be substantially *higher in winter* and *lower in summer* in the same region. This fact points to an important role of near-surface vegetation: it will less insulate from air temperature during winter and more insulate during summer with increasing climate variability in future. These effects of climate variability on thermal diffusivity of lichens and bryophytes and hence soil temperature are in the same direction as snow effects (section 3.2). Ultimately, the effects of climate variability on moss thermal diffusivity will reduce the soil warming effect of climate change in future.

Recent modelling studies suggest a soil temperature increase of 0.02 K per year since 1960 (McGuire et al., 2016) which translates into 2 K in 100 years. Such soil temperature increase has also been projected using the JSBACH model under the RCP4.5 scenario (Ekici, 2015) while under the strong warming scenario RCP8.5, the soil temperature increase might be up to 6 to 8 K (Ekici, 2015). Soil temperature differences due to climate variability differences alone (section 3.4) of the same order of magnitude (0.5-1 K) demonstrate that under increasing variability of meteorological measures and increasing extreme events in the Arctic (Seneviratne et al., 2012), the effect of gradual air temperature increase on soil temperature and hence active-layer thickness will be *substantially dampened*. Our results are conservative here because the 99 percentiles of air temperature and precipitation from the artificial dataset (REDVAR) differ by only 2-3.5 K (temperature) and 1-7 % (precipitation). These values are at the lower end of the range of climate model projections for the Arctic region until 2100 (Seneviratne et al., 2012). Such soil temperature dampening will also reduce the otherwise positive biogeochemical feedback to climate (Zimov et al., 2006; Beer, 2008; Heimann and Reichstein, 2008).

335    To address this question of validity of future soil temperature projections, for one point we additionally extended the CNTL model simulation until 2100 following the RCP 8.5. Then, in the additional experiment REDVARfut the variability of meteorological measures has been reduced continuously from 2011 until 2100 while the long-term average is still conserved. The resulting topsoil temperature difference of 0.8 K in 2100 agrees in direction and magnitude with the findings from

340    the (spatially explicit) REDVAR experiment (see above). The subsoil temperature difference is much smaller in this experiment compared to REDVAR because more time were required until the surface signal is also seen in 38 m depth.

The presented effects of short-term variability of meteorological measures on ecosystem states and functions, such as soil temperature, are also important from a methodological point of view. To

345    study the effects of environmental change on ecosystems, LSMs are usually forced by historical and reanalysis climate data for the past and present periods, and by future climate results from Earth system models. Since ESM results usually show biases, the ESM outputs cannot be used directly to drive the LSM offline model runs but first need to be bias-corrected (Hempel et al., 2013). As a result, the climate forcing shows the same trend signal as the raw ESM output but the monthly

350    mean and daily variability is corrected towards the data from the observation-based period (Piani et al., 2010; Hempel et al., 2013). Such climate forcing data can be used in general to quantify the general effects of climate change on e.g. soil temperature and permafrost thawing (Schaefer et al., 2011; Schaphoff et al., 2013; Koven et al., 2015). However, the results of the presented REDVAR and REDVARfut experiments demonstrate that using current short-term variability of climate data

355    as a proxy for future atmospheric conditions might introduce a soil temperature bias of up to 1 K.

Our findings have three major implications for future permafrost science:

1. Insulation characteristics of both snow and near-surface vegetation are responsible for a difference in soil temperature due to differences in climate variability. Therefore, future developments of LSMs should include a dynamic lichens and bryophyte model.

360    2. Statistical methods need to be developed such that future forcing data for climate change impact studies can be prepared in a way that a potential change in short-term variability and frequency of extreme events is preserved.

3. New laboratory and field experiments are required in order to confirm modelling results about climate variability effects on permafrost soil temperature.

365    In summary, projected future increases in permafrost temperature and active-layer thickness will be *less pronounced* in response to climate change than what current state-of-the-art modelling results suggest because i) current models usually do not consider dynamic functions of snow and lichens and bryophytes and ii) future changes of short-term variability of meteorological measures are not mapped to the applied bias-corrected forcing data. A first run of the MPI-ESM with the permafrost-

370    advanced land surface scheme JSBACH coupled to the atmosphere model showed a remarkable

bias in 2m air temperature of 1-4 K in permafrost regions compared to the standard model version without freezing and thawing (Hagemann et al., 2016). Our results also suggest that this bias could be potentially reduced when implementing representations of dynamic snow and of dynamic lichens and bryophytes.

**5 Conclusions**

Artificial model experiments quantify the impact of the variability of meteorological measures and their extreme events on the long-term mean of soil temperature in permafrost-affected terrestrial ecosystems. This impact is mainly due to temperature variability effects on snow melt hence snow depth as well as climate variability effects on the cover and (seasonally different) thermal diffusivity of lichens and bryophytes. Overall, the soil temperature response to increasing climate variability and extreme events (soil cooling) will be opposite to the response of soil temperature to gradually increasing air temperature (soil warming). This shows the importance of representing dynamically snow and lichen and bryophyte functions in Earth system models for projecting future permafrost soil states and land-atmosphere interactions, hence future climate. Our findings also point to the need to represent changes in short-term variability of meteorological measures in bias-corrected climate data of future periods.

*Acknowledgements.* Financial support came from the European Union FP7-ENV project PAGE21 under contract number GA282700. Model simulations were performed on resources provided by the Swedish National Infrastructure for Computing (SNIC) at Linköping University. We acknowledge the Land Department, Max Planck Institute for Meteorology, Hamburg, Germany for JSBACH code maintenance. We thank Charles Koven for a constructive review that helped to improve a previous version of the manuscript.

[Figure]

(a) Air temperature difference (K)  (b) Precipitation difference (kg m-2 s-1)

Figure 1: Comparison of 1980-2009 averages of meteorological measures (REDVAR versus CNTL).

[Figure]

(a) Air temperature standard deviation difference (b) Precipitation standard deviation difference (kg
(K)                                               m-2 s-1)

Figure 2: Comparison of 1980-2009 standard deviations of meteorological measures (REDVAR versus CNTL).

[Figure]

(a) Air temperature percentile 99 difference (K)

(b) Precipitation percentile 99 relative difference. Numbers are expressed as a fraction.

Figure 3: Comparison of 1980-2009 maximum (percentile 99) meteorological measures (REDVAR versus CNTL).

[Figure]

(a) Snow depth relative difference.

(b) Snow water equivalent relative difference.

(c) Snow density relative difference.

(d) Snow thermal diffusivity relative difference.

Figure 4: Comparison of mean winter (DJF) season snow properties during 1980-2009 (REDVAR versus CNTL). Numbers are expressed as a fraction.

[Figure]

Figure 5: Autumn (SON) 1980-2009 average snow melt relative difference (REDVAR versus CNTL). Numbers are expressed as a fraction.

[Figure]

(a) Annual lichen and bryophyte cover.

(b) Winter (DJF) lichen and bryophyte thermal diffusivity.

(c) Summer (JJA) lichen and bryophyte thermal diffusivity.

Figure 6: Comparison of lichen and bryophyte 1980-2009 average properties (REDVAR versus CNTL). Numbers are expressed as a fraction.

[Figure]

(a) Annual topsoil temperature difference (K).  (b) Annual subsoil temperature difference (K).

(c) Winter (DJF) topsoil temperature difference (d) Summer (JJA) topsoil temperature difference
(K).                                            (K).

Figure 7: Comparison of 1980-2009 average soil temperature (REDVAR versus CNTL). Topsoil and subsoil refer to depths of 3 cm and 38 m, respectively.

[Figure]

(a) Air temperature (deg C) annual mean, per- (b) Precipitation (kg/m2/s) annual mean, per-
centile 1 and percentile 99.                     centile 1 and percentile 99.

[Figure]

(c) Annual topsoil (3 cm) temperature time series (d) Annual subsoil (38 m) temperature time series
(deg C). Inset shows difference time series.        (deg C). Inset shows difference time series.

Figure 8: REDVARfut experiment results at 70N/100E during 2011-2100 showing the effects of
changing climate variability in future on soil temperature. 10-year running means are shown.

---

## Author Comment (AC4) · 12 Oct 2016

In the reply before we forgot to attach the revised version of the manuscript to which we refer to in the response. This is attached now to this reply. Christian on behalf of the co-authors.

Please also note the supplement to this comment:
http://www.the-cryosphere-discuss.net/tc-2016-210/tc-2016-210-AC4-supplement.pdf
* * *

---

## Referee Comment (RC3) · Anonymous Referee #3 · 21 Oct 2016

Reviewed paper concerns important topic of models sensitivity to input parameters. I agree with referee #1, that the paper may be more suitable for Geoscientific Model Development journal. In this research authors investigate how results of climatic models applied to ground temperature dynamics depend on forcing variability. Based on model experiments authors notice increasing of warming effect of snow and surface vegetation on ground temperature due to reduction of climate forcing variability. But if for snow cover mechanisms causing such changes were explained for vegetation cover these changes were just stated with very basic notice of water/ice content affecting moss/lichen thermal diffusivity. Investigating vegetation cover authors limit themselves with mosses and lichens as a type of vegetation has strongest influence on ground surface energy balance. But, despite of the fact that this type of surface vegetation spreads wide neglecting of other ecotypes such as grass- and shrublands, deciduous forest etc

does not allow to apply results of this study for entire northern hemisphere. Combination of lichens and bryophytes in one group was done, as far as I can understand based on similarity of its thermal properties, which is also needs to be better justified. But this approach is appropriate only for models cover short period of time. For long-term estimations differences in physiology of these plants types cannot be ignored. So due to higher productivity mosses form thick soil organic layer composed with fibrous or peat, while lichens usually associated with bare soil or very thin organic layer. Results of modelling for site located at 70N and 100E are absolutely unreal! In according to Permafrost map of Russia (http://nsidc.org/data/docs/fgdc/ggd600_russia_pf_maps/) compiled based on data collected during 1960-1980 permafrost temperature at this point was -7 to -8C and could not get colder since that time, so temperature at the depth of 38 m in 2020 cannot be so low (-13.8) as it is shown on the Figure 8d. Such a weird results make all other model outputs very doubtful. It is also not clear why authors choose this location as a reference point. Few more minor comments: Lines 200-203. It might be better to state pattern of decreasing of differences in standard deviation of variability by latitudinal zonality of diurnal temperature oscillations. Daily temperature variability in high latitudes is lower because of daylight time is extremely short during winter and extremely long during summer. Lines 288-289. I would recommend to authors be more specific and give some values of lichens/mosses thermal diffusivity obtained from their model. Lines 321-322. So low values of soil temperature increase are typical for sites where permafrost temperature now is close to the freezing point. In according to data of long-term measurements (Romanovsky, V. E., Smith, S. L., & Christiansen, H. H. (2010). Permafrost thermal state in the polar Northern Hemisphere during the international polar year 2007–2009: a synthesis. Permafrost and Periglacial Processes, 21(2), 106-116.) real permafrost temperature trends are in a range 0.05 to 0.1 K per year. Based on all abovementioned comments I cannot recommend this paper for publication as it is now. My suggestions for authors are: 1. Give better and more comprehensive description of model instead of reference to previous publication. Clarify what is a role of vegetation layer: is it a part of computation domain or boundary condition. 2. For long-term computations differences in soil organic layer formation under mosses and lichens must be taken in consideration. 3. Site-specific approach looks more appropriate for such kind of research than regional. I would recommend use database of Global Terrestrial Network – Permafrost program (http://gtnpdatabase.org/boreholes) to select sites for model validation.